# Artificial Intelligence in Digital Pathology for Bladder Cancer: Hype or Hope? A Systematic Review

**DOI:** 10.3390/cancers15184518

**Published:** 2023-09-12

**Authors:** Farbod Khoraminia, Saul Fuster, Neel Kanwal, Mitchell Olislagers, Kjersti Engan, Geert J. L. H. van Leenders, Andrew P. Stubbs, Farhan Akram, Tahlita C. M. Zuiverloon

**Affiliations:** 1Department of Urology, Erasmus MC Cancer Institute, University Medical Center Rotterdam, 3015 GD Rotterdam, The Netherlands; m.olislagers@erasmusmc.nl; 2Department of Electrical Engineering and Computer Science, University of Stavanger, 4021 Stavanger, Norway; saul.fusternavarro@uis.no (S.F.); neel.kanwal@uis.no (N.K.); kjersti.engan@uis.no (K.E.); 3Department of Pathology and Clinical Bioinformatics, Erasmus MC Cancer Institute, University Medical Center Rotterdam, 3015 GD Rotterdam, The Netherlands; g.vanleenders@erasmusmc.nl (G.J.L.H.v.L.); a.stubbs@erasmusmc.nl (A.P.S.); f.akram@erasmusmc.nl (F.A.)

**Keywords:** artificial intelligence, bladder cancer, computer-aided diagnosis, computational pathology, digital pathology, histopathology, image analysis

## Abstract

**Simple Summary:**

The diagnosis and prediction of prognosis for bladder cancer (BC) can be challenging because of the subjective nature of pathological evaluation. Artificial intelligence (AI) has emerged as a promising technology for improving the accuracy of BC diagnosis and prediction of prognosis. We reviewed all available studies that used AI to analyze images from BC tumor tissue that aimed to improve diagnosis or prediction of prognosis. Studies showed that specific tumor characteristics can be used to predict treatment response by analyzing BC tumor tissue images. Combining histopathological images with clinical information enables AI models to perform with high accuracy. In conclusion, AI has the potential to assist physicians in gaining more accurate diagnoses and treatment response predictions. Yet, important challenges should be addressed, such as ensuring reliability, interpretability, and performance—future research should address these caveats.

**Abstract:**

Bladder cancer (BC) diagnosis and prediction of prognosis are hindered by subjective pathological evaluation, which may cause misdiagnosis and under-/over-treatment. Computational pathology (CPATH) can identify clinical outcome predictors, offering an objective approach to improve prognosis. However, a systematic review of CPATH in BC literature is lacking. Therefore, we present a comprehensive overview of studies that used CPATH in BC, analyzing 33 out of 2285 identified studies. Most studies analyzed regions of interest to distinguish normal versus tumor tissue and identify tumor grade/stage and tissue types (e.g., urothelium, stroma, and muscle). The cell’s nuclear area, shape irregularity, and roundness were the most promising markers to predict recurrence and survival based on selected regions of interest, with >80% accuracy. CPATH identified molecular subtypes by detecting features, e.g., papillary structures, hyperchromatic, and pleomorphic nuclei. Combining clinicopathological and image-derived features improved recurrence and survival prediction. However, due to the lack of outcome interpretability and independent test datasets, robustness and clinical applicability could not be ensured. The current literature demonstrates that CPATH holds the potential to improve BC diagnosis and prediction of prognosis. However, more robust, interpretable, accurate models and larger datasets—representative of clinical scenarios—are needed to address artificial intelligence’s reliability, robustness, and black box challenge.

## 1. Introduction

### 1.1. Artificial Intelligence

Advancements in artificial intelligence (AI) have had a profound impact on society, particularly in the medical field [1]. AI has shown promising results in diagnostic disciplines such as radiology and pathology, providing new opportunities to analyze patient data and improve clinical outcomes [2,3]. AI systems use algorithms, which are mathematical sequences of well-defined instructions, to mimic human-like decision-making processes. The two most commonly used techniques in AI are machine learning (ML) and deep learning (DL). Both aim to enable computers to learn from data, but through different approaches (Figure 1). ML, a subfield of AI, needs a phase known as feature engineering, where specific features are manually extracted from data, particularly images, leveraging image processing techniques and human expertise (Figure 1B). This human-driven process distills relevant information from data to shape the ML model’s understanding. Next, the ML model can be trained in a supervised manner, using a set of features with known diagnosis (e.g., stage/grade) or known clinical output (e.g., recurrence/progression) to classify input data into similar groups. While ML often needs resource-intensive feature engineering and domain expertise in its design phase, DL, a subfield of ML, bypasses this by directly extracting relevant features from raw data itself (Figure 1B). DL has shown remarkable success in predicting clinical outcomes and detecting diagnostic features [4,5,6].

AI systems use different types of input data, such as clinical data, medical images, and genomic data, to learn and train their algorithms for specific applications. For example, in pathological applications, whole-slide images (WSIs) of tissue specimens can be used to train the algorithms. Two basic approaches for training algorithms are supervised and unsupervised learning [7]. Supervised learning uses labeled data as input to map data points to the label that describes them (Figure 1B). Labeled data on bladder cancer (BC) WSIs are, for example, annotating urothelium, detrusor muscle, or immune cells. Unsupervised learning does not use labeled data for training, and the algorithms operate independently to find underlying pattern clusters in the data [8].

Computational pathology (CPATH) uses different AI methods to improve diagnosis by segmentation and classification. Segmentation is the process of delineating a specific area on an image with a known pathological feature (e.g., urothelium vs. detrusor muscle), thus helping the user to identify the region of interest (ROI). Classification is identifying the features that segregate data into groups based on similarities (e.g., tumor vs. normal tissue, stage, and grade).

### 1.2. AI in BC Image Analysis

Selecting the optimal therapy for patients with BC to prevent under-/over-treatment depends on diagnostic features [9]. Current BC risk stratification systems are based on clinicopathological characteristics, but pathological evaluation suffers from intra- and interobserver variability [9,10,11,12]. Incorrect staging and grading will result in under-/over-treatment [9,10,13]. Also, increasing pathological workload and complexity highlight the need for novel tools (i.e., accurate, reproducible, fast, and affordable) to assist pathologists [14]. Extending beyond bladder-centric, several studies have showcased promising results in leveraging CPATH for improved diagnosis and prognosis prediction, with one methodology receiving FDA approval [15,16,17,18,19]. This trend is mirrored in bladder cancer, where there is currently one clinical trial focusing on computational pathology applications (NCT05825950). In BC specifically, multiple studies have shown promising results in using CPATH to improve diagnosis and prediction of prognosis through tasks like detecting tumor tissue and molecular alterations, grading, staging, and predicting clinical outcomes [4,5].

Despite the rapid development of BC CPATH, it still faces several challenges due to the complexity of analyzing and interpreting BC WSI [20,21,22]. In this review, we delve into the current status of BC CPATH. Specifically, we:Provide a comprehensive overview of the present BC CPATH landscape in diagnosis and prognosis;Highlight existing gaps between CPATH research and clinical practice;Offer recommendations to address these gaps;Discuss challenges that can shape future research in BC CPATH.

## 2. Materials and Methods

### 2.1. Literature Review

This review was conducted in accordance with the guidelines of the Preferred Reporting Items for Systematic Review and Meta-Analysis (PRISMA) [23]. Both clinical experts and algorithm developers were involved in the review process to ensure that the information presented in the studies was analyzed accurately and critically evaluated. The aim was to provide a sufficient critique of the data presented. A systematic search of English articles was performed in March 2022 using Embase, Medline, Cochrane, Web of Science, and Google Scholar. This systematic review was not registered.

### 2.2. Study Eligibility and Selection

We employed the specified terms: (bladder cancer) and (artificial intelligence) for study inclusion, along with any terms relevant or analogous to them. Detailed information on research terminology for each library and search results are provided in Appendix A. Thus, all relevant studies that analyzed BC histopathological images with an AI approach were obtained. Five exclusion criteria were applied to evaluate collected studies:i.No histopathological staining;ii.Not on bladder cancer;iii.Not using artificial intelligence;iv. No full article available;v.Non-English manuscript.

### 2.3. Data Extraction

Three reviewers (FK, SF, and NK) independently assessed the relevance of the titles and abstracts of the studies during the initial screening process. Afterward, the full texts of the remaining studies were independently reviewed to identify those that met the selection criteria. If all three reviewers agreed, a paper was selected; if they disagreed after a revision, the article was discussed with KE to decide on inclusion.

We organized the extracted data into categories, including patient and study characteristics, as well as CPATH characteristics. We extracted information on the type and number of patients, type of staining, type of images, magnification used, and performance metrics of developed algorithms, such as F1 score, accuracy, and specificity. When multiple algorithms were developed with different outcomes, we collected up to three types of outcomes with the best performance. Diagnostic performance metrics, such as sensitivity, specificity, and area under the curve (AUC), were extracted at both per-image and per-patient levels when available. We attempted to calculate missing or unclear performance metrics from the available data.

### 2.4. Data Synthesis

Given the major heterogeneity in the application of CPATH systems, study designs, algorithms, patient cohorts, evaluation strategies, and performance metrics, employing a narrative synthesis approach was deemed more appropriate than statistical pooling for our analysis. This approach allows for a detailed description and critical appraisal of the included studies, which is particularly important in diagnostic test accuracy studies where patient cohorts and test settings may differ significantly between studies and lead to biased results. Meta-analysis is not recommended in such cases. Additionally, our analysis did not include a bias assessment because of insufficient patient selection information in most studies, coupled with the current lack of a recognized reference standard for CPATH.

## 3. Results

### 3.1. Literature Search

The PRISMA flow diagram (Figure 2) provides a graphical representation of the systematic search conducted in this study. After the literature search, 2285 studies were included, of which 720 were removed since they were duplicated. The remaining 1565 documents were independently assessed by FK, SF, and NK, based on the title abstract, and 1398 studies were excluded after reaching an agreement. One study could not be retrieved, 166 studies were independently assessed based on the full text, and 135 studies were excluded according to the exclusion criteria. Additionally, two studies were included based on the snowball search. In the final analysis, 33 studies were included. Table 1 provides a comprehensive summary of patient characteristics, study aims, and the methods used.

### 3.2. Applications of AI Methods for Diagnosis

CPATH systems use WSIs to segment tissue types (e.g., urothelium, connective tissue, or muscle) and/or identify cell characteristics (e.g., nuclei, cytoplasm) [24,25,26]. Segmented tissue can be used for classification, for example, normal vs. tumor tissue, grading, or staging. Below, we discuss the current knowledge of BC WSI segmentation in more detail.

#### 3.2.1. Tissue and Cell Segmentation

CPATH can accurately detect diagnostically relevant areas at both tissue and cellular levels from BC WSI. Automated detection of the relevant regions can accelerate pathological assessment by directing the pathologist and increasing diagnosis accuracy [27].

Niazi et al. developed an algorithm that segmented lamina propria, muscularis propria, and urothelium with 98%, 98%, and 94% accuracy, respectively [24]. In the same line, Wetteland et al. developed an algorithm that segmented urothelium, stroma, muscle, blood, damaged tissue, and background with 96% average accuracy [28]. They showed that analysis based on different magnification levels improved the segmentation accuracy.

Segmentation algorithms also showed promising results for segmenting microvessels and cell nuclei characteristics [26,29]. Neovascularization indicates metastatic potential, and intratumoral microvessel density can provide diagnostic information [30]. Loukas developed a model using immunohistochemistry-stained images that segmented microvessels with 87% accuracy [29]. Certain characteristics of cell nuclei, such as alterations in polarization and shape (e.g., loss of roundness), have been found to correlate with worse clinical outcomes [29]. Nevertheless, analyzing these characteristics is time-consuming and operator-dependent. Therefore, computationally assessing cell nuclei characteristics has the potential to improve the prediction of prognosis. Glotsos et al. used BC biopsy images to develop a cell nuclei segmentation model that reached 94% accuracy [26]. Unfortunately, clinical outcome was not investigated in these studies.

#### 3.2.2. Detection of Tumor vs. Normal Tissue

Accurate identification of tumor areas within a WSI can improve diagnosis by guiding the pathologist to a specific ROI, hereby reducing assessment time and intra-/inter-observer variability [27]. Zhang et al. developed an algorithm to discriminate tumor and normal areas using WSIs from patients with papillary urothelial carcinoma [5]. Two pathologists annotated tumor and normal tissue areas to train the algorithm. The algorithm retrieved the tumor and normal urothelium with a 0.95 true positive rate. Despite the large dataset in this study, the developed method only focused on papillary tumors; tumor types such as carcinoma in situ (a flat lesion), which might be more difficult to detect, were not included.

Noorbakhsh and colleagues used WSIs of 19 tissue types to develop an algorithm to classify tumor vs. normal images, which reached an area under the curve (AUC) of 0.99 [31]. The algorithm trained on BC, invasive breast carcinoma, and endometrial carcinoma could correctly classify tumor and normal tissue areas in other organ types (with AUC of 0.98, 0.97, and 0.97, respectively). In the same line, Jang et al. used WSIs from bladder, lung, colon, rectum, stomach, bile canal, and liver tissue to develop an algorithm that classified normal vs. tumor areas with 0.94–0.98 AUC on BC WSIs (models trained on five different cancer types other than BC) [32]. Therefore, image datasets from various cancer types could be used to train an algorithm operated on another tissue type based on morphological similarities.

For decision assistance, content-based image retrieval (CBIR) systems retrieve comparable images with associated information, such as a pathology report or clinical result. CBIR, by recalling previously diagnosed images/reports, can aid pathologists in identifying similar images and increase diagnostic accuracy. Kalra and colleagues trained an algorithm with WSIs from 32 tumor types other than BC to retrieve similar cancer-type images, achieving 96% accuracy [33]. Khosravi et al. also developed a method using WSIs with bladder, breast, or lung cancer as a general label to classify cancer types, achieving 100% accuracy [34]. These findings suggest that by looking at a new case, CPATH can retrieve similar cases diagnosed previously.

#### 3.2.3. Grading and Staging

BC diagnosis and current risk stratification systems mainly rely on clinicopathological factors such as tumor grading and staging [9,10,35]. Incorrect grading and staging can lead to under-/over-treatment. CPATH has the potential to improve grading and staging diagnosis by providing pathologists with an accurate and reproducible second opinion [36,37].

Yin et al. developed an algorithm that distinguished Ta vs. T1 disease with 96% accuracy [37]. Extracted features from images such as nuclear size, cytoplasmic color, nuclear shape, and the pattern of connective tissue around the tumor were used to train the ML model. An in-depth analysis of the results showed that desmoplastic reaction was the most important feature in distinguishing Ta from T1 tumors. A limitation of this study was the exclusion of challenging cases to distinguish Ta vs. T1 for pathologists, which makes the dataset unrepresentative of real clinical practice.

Cancer grading relies on cell nuclei features, and automated identification of cell nuclei could assist pathologists in improving the grading assessment [38,39,40]. Spyridonos et al. developed an algorithm that segmented cell nuclei and, by using ML methods, 36 extracted nuclei features from annotated regions. This algorithm achieved 88% in classifying tumor areas into high and low grades [39]. The same group also developed an algorithm based on the WHO 1973 grading system, which classified selected tumor areas into grades 1, 2, and 3 with an efficiency of 89% in distinguishing grade 1 and 2 cases from grade 3, and 79% among grade 1 vs. grade 2 cases [40]. In a later study, they developed a grading algorithm using non-muscle-invasive BC WSIs that achieved 85% accuracy [41].

Zhang and colleagues trained a method to identify tumor areas and provide grades for these detected areas [5]. The automated results were compared to assessments conducted by a panel of 17 pathologists performing the same tasks. The developed method outperformed the pathologists in grading, achieving 95% accuracy compared to the pathologists’ 84% accuracy. The ground truth was based on grading provided by four independent pathologists. Papageorgiou et al. developed a grading algorithm that achieved 89% and 86% accuracy for high-grade and low-grade tumor areas, respectively [42].

Wetteland et al. developed an algorithm using general grading labels for each WSI that correctly graded 90% of the WSIs [43]. Different tissue types were automatically segmented [28], and urothelium was used for grading analysis. Along the same line, Jansen and colleagues developed two methods to segment urothelium and provide grading, achieving 76% accuracy for low-grade and 71% accuracy for high-grade cancerous urothelium in WSIs from non-muscle-invasive bladder cancer (NMIBC) patients [44].

Muscle Invasive Bladder Cancer (MIBC) patients have different histological patterns associated with varying prognoses and aggressiveness [45]. Jimenez et al. identified three invasive growth patterns that can predict recurrence in MIBC patients: nodular, trabecular, and infiltrative [46]. Garcia et al. developed a method using immunohistochemistry-stained WSIs to classify normal, infiltrative, nodular, and trabecular histological patterns, achieving an average accuracy of 90% [47].

#### 3.2.4. Generation of a Diagnostic Report

CPATH has shown promising results in tissue segmentation, grading, and staging. However, the CPATH outcome is often challenging to interpret, especially for DL-based models, due to the lack of transparency in the decision-making process, often referred to as the “black box”. Clinicians cannot make healthcare decisions based on the “black box” unless they can rely on the outcome of a method. Having accurate and reproducible methods or comprehending the findings’ rationale can shed light on the black box. Therefore, translating CPATH results into clinically relevant output (e.g., a diagnostic report) is needed.

Zhang et al. developed a set of algorithms based on pathological reports written for image patches to describe morphological features [48]. These trained algorithms generated pathological reports and retrieved the correlated images with 79% accuracy—outperforming well-known image-to-text-retrieval methods in the same task. In a subsequent study focused on image-to-text retrieval, two pathologists briefly characterized the cell features of 4253 split images, and four pathologists transliterated the corresponding reports [5]. An image-to-text retrieval algorithm was designed to generate diagnostic reports, and a text-to-image retrieval method was built to visualize the image pixels responsible for the reported findings, thereby enhancing the interpretability of the outcome. Both methods outperformed state-of-the-art techniques in report generation. Explaining the CPATH outcome can shed light on the black box and fill the gap concerning CPATH methods and clinical practice. Nevertheless, CPATH systems should work together with pathologists in the future and not stand alone.

### 3.3. Applications of AI Methods for Prognosis

#### 3.3.1. Predicting Clinical Outcome

Approximately 30–50% of BC patients experience recurrence or progression after treatment, and predicting clinical outcomes is essential to select the appropriate treatment [9,10,49,50]. The existing prediction methods rely on clinicopathological data, which are often insufficient, thus emphasizing the need for an affordable, rapid, reproducible, and effective technique. CPATH has the potential to predict clinical outcomes by analyzing tumor growth patterns, cell nuclei, and microenvironments [51,52,53,54,55].

Tumor budding (TB) is represented by a cluster of up to four cancer cells located at the invasive front of a tumor that correlates with poor prognosis in many cancer types [56,57]. Brieu et al. developed a method to quantify TB using MIBC WSIs, demonstrating a correlation between TB and poor survival [51]. They showed that combining TB-detected features with clinical parameters improved the clinicopathological-based prognostic approach by separating MIBC patients into low and high TB groups associated with disease-specific survival.

Tumor cell nuclei undergo significant changes, and when quantified, these modifications can diagnose cancer or predict the disease’s course [58]. ML-based CPATH methodologies that evaluate cellular features show promising potential in predicting cancer recurrence and survival [52,53,54,55]. The mutually analyzed cellular features were mostly cell nuclear area, skewness of area, and circularity. Tasoulis et al. investigated the quantitative descriptors of nuclear morphometry (e.g., maximum area, skewness of area, maximum concavity) to predict recurrence, which achieved an accuracy of 92% [52]. Chen et al. developed an algorithm to predict overall survival (OS) [53]. Quantitative extracted phenotypic tissue features of object shape, size, and texture from WSIs (e.g., cell nuclei area, contrast, and distribution) were combined with clinical information. The developed algorithm in this study, using combined data, achieved an 81% accuracy in predicting 5-year overall survival (OS) and outperformed current risk stratification systems based on clinicopathological characteristics. In another study, the same group developed an algorithm using BC WSIs from patients who underwent a radical cystectomy to predict OS by extracting and analyzing phenotypic features (e.g., nucleus/cell area ratio, nucleus circularity, and cell area) [54]. Extracted features were merged with the neutrophil-to-lymphocyte ratio obtained from peripheral blood or estimated from transcriptomic data. Their novel nomogram could predict OS and perform better than tumor grade and stage systems in decision curve analysis.

Tokuyama and colleagues developed an algorithm using NMIBC WSIs to predict recurrence using extracted nuclear features from pathologist-annotated ROIs (e.g., area, mean radius, correlation), which reached 90% accuracy [55]. Gavriel et al. used multiple ML algorithms to predict cancer-specific survival (CSS) from immunofluorescence-stained MIBC WSIs [59]. Firstly, tumor budding, T-cells, macrophages, and their co-expression of the immune checkpoint ligand PD-L1 were identified. Subsequently, spatial and image features (e.g., the number and density of different cell types and the total tumor area) were identified and combined with clinical information to predict CSS. This model had 89% AUC and 80% accuracy for predicting CSS, outperforming the current clinicopathological-based model. Mi et al. developed an algorithm based on measured size and angles of cell nuclei that predicted response to neoadjuvant chemotherapy in MIBC patients with 73% accuracy [60].

Some studies used a DL approach to predict recurrence and lymph node metastases at radical cystectomy [6,61]. Lucas et al. trained an algorithm to predict 1- and 5-year recurrence-free survival that reached AUCs of 0.62 and 0.76, respectively—combining image features with clinical data improved recurrence prediction [6]. Harmon et al. developed a method to predict lymph node metastasis (LNM) for MIBC patients at radical cystectomy [61]. Saltz et al. extracted image features from annotated tumor areas were used to develop a multi-magnification model to show the probability map of LNM, and the result was combined with spatial tumor-infiltrating-lymphocytes (TILs) probability [62]. Subsequently, the output was combined with a clinicopathological-based logistic regression model that achieved an AUC of 0.80, higher than the clinicopathological (0.67) or ML (0.78) models alone.

#### 3.3.2. Detection of Biomarkers and Molecular Alterations

Molecular alterations such as fibroblast growth factor receptor (FGFR) and HER2 have been shown to be associated with response to treatment and clinical outcome, suggesting them as predictive factors [63,64]. Nevertheless, detecting molecular alterations in clinical practice is hindered by high costs and complexity, highlighting the need for alternative, affordable, and rapid methods. CPATH methods have shown promising results in detecting molecular alterations and are relatively cheap and fast [4,25,31,34,65]. Therefore, CPATH can be used as a pre-selection tool to determine eligibility for molecular testing and, ultimately, help with bridging the gap between molecular identification techniques and clinical practice.

MIBC can be classified into distinct prognostic and predictive molecular subtypes: luminal, basal squamous, neuronal, and stroma-rich [66,67,68]. Woerl et al. developed an algorithm to predict molecular subtypes from MIBC WSIs that reached 75% accuracy [4]. Pathological interpretations were made on 800 randomly selected tiles from detected areas by the trained algorithm. The following cellular and histological structures, hyperchromatic nuclei with low to moderate pleomorphism; large, pleomorphic nuclei with multiple atypical nucleoli; papillary structure; small cellular nests or diffusely infiltrating single tumor cells were identified as most relevant in double negative, basal, luminal, and luminal p53-like molecular subtypes respectively. Four pathologists were presented with the algorithm’s detected areas for each molecular subtype, and their accuracy in predicting molecular subtypes increased from 38% to 59%.

Noorbakhsh et al. developed an algorithm to detect TP53 mutations using WSIs from various cancer types, including BC, breast cancer, lung adenocarcinoma, stomach adenocarcinoma, and colon cancer. The algorithm achieved an AUC of 0.7 to detect TP53 mutation from BC WSIs [31]. Similarly, Khosravi et al. developed an algorithm that identified four BC-specific biomarkers (Table 1) with accuracies of 99% and 83% on two datasets [34].

The expression of Ki-67 (Ki-67 index) is a proliferation marker, and a high expression correlates with poor clinical outcomes. Lakshmi et al. developed a method for cell nuclei segmentation and classification to compute the Ki-67 index [65]. Cell nuclei were segmented with an average accuracy of 93%, and the Ki-67 index was calculated with a margin error of 2.1%. Manual labeling of cell nuclei to train the algorithm was a limitation of this study, addressed in a subsequent study by the same group through the use of automatically labeled data [25]. Their improved algorithm achieved an 89% accuracy in segmenting cell nuclei, with the Ki-67 index measured with a 0.4% error.

TIL density is correlated with clinical outcomes, such as prolonged disease-free survival or increased OS, across various cancer types. Saltz et al. developed an algorithm using WSIs from 13 cancer types, including BC, to detect TILs [62]. To boost the performance, they developed an algorithm to learn the representation of nuclei and lymphocytes. Then, the enhanced algorithm was trained and optimized with pathologist-labeled images. The developed model reached 0.95 AUC to detect TILs.

FGFR mutations are frequently detected in BC and can serve as prognostic markers for treatment response [69]. Velmahos et al. developed a method to predict FGFR-activating mutations [70]. A higher frequency of FGFR mutations correlates with a high TIL percentage and vice versa [71]. Accordingly, the TILs proportion was estimated (using Saltz et al.’s method [62]) to evaluate FGFR mutations, and the algorithm predicted FGFR mutation with 0.82 sensitivity and 0.42 specificity. Loeffler et al. developed an algorithm to detect FGFR3 mutations that reached 0.72 AUC [72]. In summary, CPATH can potentially translate molecular alterations-based prognostic models into clinical practice.

In light of our findings, we have identified key gaps in BC CPATH and provided actionable recommendations along with their underlying reasons. Due to the variations in research questions, validation methods, and the WSIs and corresponding annotation databases used, a direct comparison of the studies was not feasible. Nevertheless, Table 2 provides a guide to address the existing challenges and effectively incorporate CPATH into the field of BC diagnosis and prognosis.

**Table 1 cancers-15-04518-t001:** Overview of included studies.

Study	Year	Aim of the Study (Related to AI Image Analysis on BC)	Diagnosis or Prognosis	Dataset	Number of Patients	Type of Model in Use	Staining Type
Niazi et al. [24]	2020	Tissue type segmentation (urothelium, stroma, muscle, and blood)	Diagnosis	In-house	54 (T1 samples)	Supervised DL	HE
Wetteland et al. [28]	2020	Tissue type segmentation (urothelium, stroma, muscle, and blood)	Diagnosis	In-house	39	Supervised DL	HES
Loukas [29]	2013	Vessel segmentation	Diagnosis	In-house	107	Unsupervised ML	CD31
Glotsos et al. [26]	2004	Cell nuclei segmentation (for selected urothelium regions)	Diagnosis	In-house	50	Supervised ML	HE
Zhang et al. [5]	2019	Detecting tumor area, grading classification, producing an interpretable pathology report	Diagnosis	TCGA, in-house	913	Supervised DL	HE
Jang et al. [32]	2021	Tissue classification into tumor vs. normal areas for six cancer types to assess the generalizability of diagnostic DL models.	Diagnosis	TCGA	NA	Supervised DL	HE
Kalra et al. [33]	2020	Pan-cancer classification with CBIR approach to assess diagnostic consensus through searching archival WSIs	Diagnosis	TCGA	410 (before exclusion)	Unsupervised DL	HE
Yin et al. [37]	2020	Ta and T1 staging classification	Diagnosis	In-house	1177	Supervised ML	HE
Spyridonos et al. [39]	2001	Nuclei segmentation and tumor grading	Diagnosis	In-house	92	Supervised ML	HE
Spyridonos et al. [40]	2002	Nuclei segmentation and tumor grading	Diagnosis	In-house	92	Supervised ML	HE
Spyridonos et al. [41]	2006	Tumor grading	Diagnosis	In-house	129 (NMIBC patients)	Supervised ML	HE
Papageorgiou et al. [42]	2006	Tumor grading	Diagnosis	In-house	129	Unsupervised and supervised ML	HE
Wetteland et al. [43]	2021	Tumor grading	Diagnosis	In-house	300 (NMIBC patients)	Supervised DL	HE
Jansen et al. [44]	2020	Automated tumor detection and grading	Diagnosis	In-house	232 (NMIBC patients)	Supervised DL	HE
García et al. [47]	2021	Detecting histological patterns (normal, mild, or trabecular in IHC images	Diagnosis	In-house	136	Unsupervised DL	IHC (Cytokeratin AE1/AE3)
Zhang et al. [48]	2017	Produce an interpretable pathology report for the corresponding ROI	Diagnosis	TCGA, in-house	32	Supervised DL	HE
Noorbakhsh et al. [31]	2020	Classifying tumor/non-tumor slides, cancer subtype, and TP53 mutation	Diagnosis and prognosis	TCGA	27,815 ^a^	Unsupervised DL	HE
Khosravi et al. [34]	2018	Tissue type (bladder, breast, and lung cancer) and biomarker classification	Diagnosis and prognosis	TCGA and TMAD	2139 IHC, 543 H&E, and 2139 IHC images	Supervised DL	HE, IHC (CK14, GATA3, S0084, and S100P)
Brieu et al. [51]	2019	Detecting tumor budding to improve prognosis by predicting survival	prognosis	In-house	100	Supervised DL and ML	HE
Tasoulis et al. [52]	2006	Collecting and quantification of cell nuclei characteristics to improve prognosis by predicting recurrence	prognosis	In-house	127	Supervised ML	HE
Chen et al. [53]	2021	Predicting overall survival by using extracted quantitative phenotypic tissue features	prognosis	TCGA and in-house	514	Supervised ML	HE
Chen et al. [54]	2021	Provide a novel nomogram for decision-making and predicting overall survival by using extracted quantitative features and combining them with neutrophil-to-lymphocyte ratio information	prognosis	TCGA and in-house	508	Supervised ML	HE
Tokuyama et al. [55]	2021	Predict recurrence by using extracted quantitative nuclei features	prognosis	In-house	125 (NMIBC patients)	Supervised ML	HE
Gavriel et al. [59]	2021	Predict cancer-specific survival by combining image, clinical and spatial features extracted from IHC images	prognosis	In-house	78	Supervised ML	IHC (Pan CK, CD3, CD8, CD68, CD163, PD-L1)
Mi et al. [60]	2021	Predict response to neoadjuvant chemotherapy by using extracted cell nuclei features	prognosis	TCGA and in-house	73	Supervised DL	HE, IHC (P16, P53, P63, Ki67, CK20, CK5/6, GATA3, and Her2Neu)
Lucas et al. [6]	2020	Predicting recurrence by combining image features with clinical information	prognosis	In-house	359	Unsupervised DL	HE
Harmon et al. [61]	2020	Predicting lymph node metastasis by combining extracted image features with a spatial tumor-infiltrating-lymphocytes probability model	prognosis	TCGA, in-house	307	Supervised DL and ML	HE
Woerl et al. [4]	2019	Predicting molecular subtypes from H&E slides	prognosis	TCGA, in-house	379	Supervised DL	HE
Lakshmi et al. [65]	2019	Estimating Ki-67 index by segmentation and classification of cell nuclei	prognosis	In-house	8 ^b^	Supervised DL	Ki-67
Lakshmi et al. [25]	2020	Estimating Ki-67 index by segmentation and classification of cell nuclei, which use automatically labeled data	prognosis	In-house	8 ^b^	Supervised DL	HE
Saltz et al. [62]	2018	Mapping of tumor-infiltrating lymphocytes by training an algorithm that shows the representation of cell nuclei and lymphocytes, which is optimized with pathologist-labeled data	prognosis	TCGA	5202 ^c^	Supervised DL	HE
Velmahos et al. [70]	2021	Predicting FGFR mutation by estimating TILs proportion	prognosis	TCGA	290	Supervised DL	HE
Loeffler et al. [72]	2021	Detecting FGFR3 mutation from H&E images	prognosis	TCGA and in-house	574	Supervised DL	HE

BC—bladder cancer; TCGA—the cancer genomic atlas; TMAD—the Stanford tissue microarray database; AI—artificial intelligence; ML—machine learning; DL—deep learning; HE—hematoxylin and eosin; HES—hematoxylin eosin saffron; IHC—immunohistochemistry; CD31—a cluster of differentiation 31; NMIBC—non-muscle-invasive bladder cancer; FGFR—fibroblast growth factor receptor; CBIR—content-based image retrieval; ROI—regions of interest; ^a^—flash-frozen samples from 19 cancer types; ^b^—80 images from 8 patients; ^c^—from 13 cancer types.

**Table 2 cancers-15-04518-t002:** Identified areas to improve and proposed recommendations for future research to integrate CPATH into BC clinical practice.

Research Phase	Area to Improve	Recommendation	Reason
Data collection	Model robustness	Large number of patients (>100)	Avoid overfitting and develop accurate models
	Open access resources	Publish WSI and annotation dataset publicly	Reproducible output
	Patient monitoring period	Period-covering follow-up	Cover the full timeframe to assess treatment efficacy
	Process uniformity	Standardization of data collection and keeping records of each step	Ensure consistency, reproducibility, and increase transparency for legal aspects
Data pre-processing	Image dataset quality	Remove noise, such as artifacts, from the images	Increase generalizability and accuracy
Experiments and analysis	Study design transparency	Keep track of collection and adjustments made in the dataset, experiments, and algorithm	Reproducible study design and increase transparency for legal aspects
	Transparent algorithm design	Publish the developed algorithm publicly	Reproducible output
	Results consistency and standardization	Report all basic results for classifications (e.g., accuracy, F1 score, AUC)	Make results comparable
	Cross-demographic algorithm evaluation	Further validation and testing of CPATH algorithms in diverse patient populations	ensure generalizability and accuracy
Interpretation	Transparency in the decision-making process	Assess the outcome and interpret the rationale behind the decision an algorithm has made	Shed light on the black box for transparent and legally acceptable outcomes in clinical practice
Clinical implementation	Clinical utility and efficacy assessment	Implement trained models in the clinical setting	Integrating the AI models into the clinical workflow
	Adaptive learning	Monitor the model’s performance and update it with new data	Maximize the model’s utility in real-world settings in varied scenarios

## 4. Discussion

In this review, we provide a comprehensive overview of CPATH’s role in improving BC diagnosis and prediction of prognosis. CPATH has shown promising results in enhancing diagnostic and prognostic prediction accuracy, paving the way for personalized treatment decisions. Nevertheless, because of variability in performance metrics, datasets in use, and methodologies across studies, direct comparison of studies was not feasible. This highlights the need for a standardized framework, as we recommended in Table 2, to facilitate the integration of research findings, thereby ensuring comparability and reproducibility across studies.

### 4.1. CPATH for BC Diagnosis

In the BC diagnostic context, CPATH mainly targets the detection of ROIs (such as tumors, cell nuclei, and tissue types), the evaluation of their grading and staging, and the production of diagnostic summaries. To improve the diagnostic CPATH systems’ efficiency, they can focus on automatically segmented areas with diagnostic relevance, and areas that diagnostically are not relevant, such as damaged tissue and blood, can be excluded [73]. For instance, segmented urothelium can be used as input to develop a CPATH algorithm that aims to provide grading. On the topic of segmentation, three studies achieved remarkable accuracy, surpassing 89%, for segmenting urothelium and cell nuclei using CPATH [24,26,28]. Seven studies that used ML and DL-based CPATH models achieved high grading accuracy (85%–96% for ML-based analysis of cell nuclei features and 74%–95% for DL-based models) [5,39,40,41,42,43,44]. However, the majority of these studies focused primarily on carefully selected ROIs for analysis. This approach may overlook vital but less conspicuous diagnostic information elsewhere in the image [74,75]. Additionally, it could lead to models that are less robust due to their dependency on specific regions, thereby undermining the full potential of CPATH methods. To circumvent this bias and ensure transferability to clinical practice, all areas on the slide need to be analyzed. However, some areas, such as those containing artifacts or diagnostically irrelevant features, can compromise the model’s reproducibility and accuracy. Therefore, excluding these areas can enhance both reproducibility and accuracy, fostering the development of CPATH methods with less human intervention. Considering these conditions, Table 2 offers recommendations to mitigate selection bias and promote the clinical transferability of CPATH methods.

### 4.2. CPATH for BC Prediction of Prognosis

Prediction of prognosis in BC is a crucial aspect of patient care, guiding treatment plans and decisions. Currently, clinicopathological factors are the primary determinants used in clinics, yet their disease course prediction remains suboptimal. Incorporating CPATH can enhance prognostic accuracy and pave the way for exploring and validating newer, previously less-investigated biomarkers in BC. For example, one study demonstrated desmoplastic reaction as a leading indicator for staging [37]. Although well-established in other types of cancers, such as gallbladder and colorectal, the role of desmoplastic reaction presents a ripe area for deeper exploration in BC [76,77]. The mentioned study that highlighted the importance of desmoplastic reaction in BC used an ML-based method to differentiate between Ta and T1 stages by analyzing H&E WSIs, which reached 96% accuracy [37]. By tracing back to the influential features in their ML model, desmoplastic reaction was identified as the leading indicator for staging.

On a similar note, mitotic figures are another prognostic factor in several cancers, such as breast, bladder, and melanoma [78,79,80], with their frequency serving as a determinant in distinguishing low- from high-grade BC and predicting recurrence [79]. In various cancer types, including breast cancer and melanocyte lesions, mitosis has been successfully detected using CPATH methods [81,82]. Although one study used automated mitosis detection for objective grading in BC, there is more untapped potential in this field [83]. The automated highlighting of mitotic figures in BC WSIs has the potential to decrease inter- or intraobserver variability in diagnosis. Moreover, detecting and quantifying mitotic figures can improve the prediction of prognosis; one included study used a similar approach to enhance the prediction of prognosis through the automated detection of tumor budding [51]. This finding suggests the potential of CPATH in unraveling prognostic and diagnostic markers in BC.

Segmented cell nuclei features have shown predictive capabilities for prognosis in glioma, renal cell carcinoma, and lung cancer [84,85]. Building on these findings, five included studies explored the use of ML-based models and analyzed extracted cell nuclei features to predict the prognosis of BC [52,53,54,55,59]. These models achieved minimum accuracies of 90%, 81%, and 80% for recurrence, overall survival, and cancer-specific survival, respectively. A key advantage of these studies was their ability to trace back and identify predictive variables within their models, demonstrating the strength of ML-based methods in prognostic marker identification. Two studies employed DL methods to predict recurrence and lymph node metastasis, achieving AUCs of 0.62 and 0.8, respectively. However, the limited interpretability of DL models hinders the ability to recognize specific features that contribute to outcome prediction. The study on predicting lymph node metastasis showed that an integrated analysis of histopathological images and clinicopathological data can improve the prediction. Moreover, a notable study investigating glioma and clear cell renal cell carcinoma underscored the use of a multimodal fusion paradigm [85]. This approach enriches prognostic predictions by synergistically analyzing genomic features alongside morphological characteristics extracted from histopathological images. It offers persuasive evidence that future BC studies on the prediction of prognosis can be improved by adopting such multimodal strategies. For instance, integrating clinicopathological information, histopathological imagery, and molecular subtyping data can potentially open doors to superior predictive outcomes.

CPATH’s versatility in prognostic tasks extends further; for example, CPATH studies have demonstrated the promising capability of predicting molecular subtypes from WSIs in breast cancer [86]. In a notable included study on BC that used a DL-based method, molecular subtypes were predicted with 75% accuracy and 0.89 AUC [4]. While the performance level has not reached clinical practice standards, it remains significant given the tumor heterogeneity in BC, where diverse genetic and phenotypic profiles within one tumor can challenge classifications. Their method’s ability to detect molecular subtypes from histopathological images without intensive molecular assessment can pave the way for more efficient diagnostic methods. Moreover, the importance of detecting molecular subtypes is heightened by their clinical relevance. Furthermore, by evaluating randomly selected influential areas for the algorithm’s decision, a correlation between distinct molecular subtypes and specific morphological features was found—a novel finding. Thus, when using DL models, underscoring image areas influencing algorithm decisions improves transparency and interpretability, secures reliable results, and simplifies decision-making comprehension. As a result, users can ascertain accuracy, identify biases, improve the performance and reliability of the methods, and even unlock the potential to discover novel prognostic features.

### 4.3. Navigating the Future: Challenges and Improvements in BC CPATH

The above-mentioned findings underscore the effectiveness of CPATH models in tasks such as predicting prognosis, grading, and segmenting ROIs. Such use of CPATH has the potential to streamline pathologists’ workflows by guiding their attention towards significant areas, like tumors, or provide grading. Pathologists’ workload is increasing, given factors such as population growth, advancements in medical technology, and evolving diagnostic criteria [14]. In several cancer types, such as prostate, colorectal, and breast, the potential of CPATH in improving diagnosis and prediction of prognosis has been proven by multiple notable studies, with one of the methodologies getting FDA approval [15,16,17,18,19]. In BC, this trend is reflected in a present clinical trial focusing on the CPATH approach (NCT05825950). Moreover, in a recent notable study, Wu et al. developed an AI-based model to diagnose lymph node metastases in BC from WSIs, which showed superior diagnostic sensitivity over both newer and experienced pathologists. Impressively, its diagnostic ability wasn’t limited to BC but extended to breast and prostate cancers as well. With this AI breakthrough, pathologists are better equipped to identify hard-to-spot micrometastases that could otherwise be overlooked [87]. However, AI-based models are not designed to replace human expertise—but to augment the expertise of clinicians. Consequently, there is a need to foster collaboration between AI system creators and clinicians to co-design the CPATH methods. Moreover, clinicians cannot make decisions based on a black box. Thus, enhancing AI model transparency through feature tracing becomes essential, fostering trust, accountability, and explainability in AI-driven decision support systems. Additionally, ethical and legal implications such as patients’ privacy, informed consent, data protection, and fair use of the technology must be considered. Establishing ethical and legal frameworks can ensure the responsible design and use of AI-based CPATH.

CPATH’s efficacy is challenged by the need for large, varied datasets, standardized data practices, and interpretable models for clinical endorsement. The widespread use of The Cancer Genome Atlas (TCGA) dataset, as seen in 42% of the studies included in our review, further underscores the importance of addressing dataset biases. Some models trained on TCGA have shown a tendency to recognize specific institutional patterns, which, although not medically relevant, could unintentionally affect model performance [88,89]. Moreover, the lack of cross-validation among different cohorts, potential lab-induced tissue artifacts, and the biases from institutional patterns limit model generalizability and clinical application. Elevating the quality and functionality of CPATH in BC demands substantial clinical datasets with adequate clinical follow-up, image noise removal, standardization, and tracking of data collection. Validating models on a broad range of datasets outside of TCGA is essential in upcoming studies to minimize potential biases. Public release of both WSI and annotation datasets, alongside comprehensive disclosure of the algorithm’s outcome and clear decision justifications, remains essential.

To maximize the potential of AI in enhancing patient outcomes, it’s essential not just to develop CPATH models but to effectively integrate them within clinical workflows. Continuous monitoring and updating of these models are essential for sustaining their accuracy and reliability. One of the emerging innovations that can help with integrating CPATH in BC clinical practice is fusion models [85]. These models can offer enhanced prediction of prognosis by combining multiple AI techniques and acquiring the unique strengths of individual algorithms.

Patient stratification in clinical trials is essential for their success, which can be augmented by using CPATH’s predictive capabilities. By leveraging CPATH, we can better predict which patients will benefit from specific treatments, thus refining precision medicine and enhancing patient outcomes. An example clinical trial can be more efficient by identifying HR-NMIBC patients who are unlikely to benefit from BCG treatment, allowing for their enrollment in trials investigating other potential treatments; similarly, a subset of MIBC patients who might benefit from immune checkpoints can be identified. CPATH can pinpoint these patient groups. Additionally, the adoption of explainable AI offers transparency into algorithmic decision-making, facilitating clinicians’ understanding and confidence in these tools. Importantly, the efficacy of CPATH models depends on diverse datasets, ensuring adaptability across patient groups. Thus, a comprehensive data foundation enhances personalized care for each individual. Beyond its technical innovation, CPATH is directing the BC field towards a paradigm shift in patient care—when CPATH meets BC, hope prevails over hype.

## 5. Conclusions

In conclusion, CPATH holds the potential to improve BC diagnosis and prediction of prognosis. By using ML and DL algorithms, CPATH can detect ROIs, grade and stage tumors, and predict clinical outcomes with high accuracy. Importantly, these AI-based models are not intended to replace pathologists but to augment their expertise, helping them to make more informed decisions. However, implementing CPATH effectively in clinical practice requires addressing several challenges. These challenges include standardization of data collection and analysis, interpretation and explainability of AI models, validation of diverse patient cohorts, and ethical and legal implications. To improve the reliability and usefulness of CPATH research, it is important to establish legal frameworks, implement adequate clinical follow-up and standardize data collection, publish datasets publicly, and ensure transparency and ethical use of AI-based tools.

Despite these challenges, CPATH has the potential to revolutionize BC management by identifying novel biomarkers and therapeutic targets and facilitating the development of more effective treatment strategies. By integrating histological data with clinical and molecular data, CPATH can provide a more comprehensive understanding of the disease and its underlying mechanisms, leading to precision medicine and improved patient outcomes. Thus, AI-based histopathological image analysis is a hope, not a hype, for BC clinical practice.

## Figures and Tables

**Figure 1 cancers-15-04518-f001:**
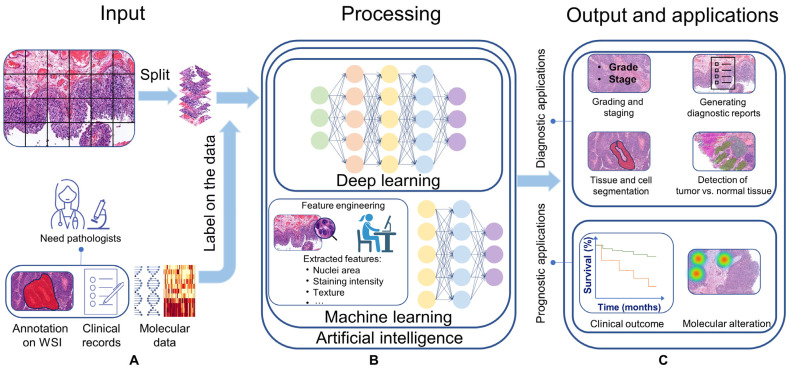
A general overview of types of input, processing methods, output, and applications of AI systems in analyzing H&E WSIs. (**A**) Input for the algorithm are the image tiles that are extracted from WSIs and can be labeled for processing and output evaluation. Labeled data can include clinical records, molecular data, and annotations like stage, grade, and tissue types. (**B**) AI, ML, and DL can be used to process input data. ML is a subfield of AI, and DL is a subfield of ML. ML algorithms learn from hand-crafted features and require feature engineering in the design phase to extract a set of features and train the algorithm. DL algorithms automatically extract relevant features from raw data (without human intervention) to train the algorithm and stratify features into similar groups via classification networks. (**C**) Output and applications of CPATH in BC image analysis can assist pathologists in improving the diagnostic process, predicting clinical outcomes, or discovering novel features. Diagnostic applications mainly focus on tissue and cell segmentation, tumor vs. normal tissue detection, grading and staging, and generation of a diagnostic report for clinical implementation. Current prognostic applications focus on predicting clinical outcomes or molecular alterations. The output can be evaluated using labels on the dataset. Abbreviations: WSI—Whole-slide image, AI—artificial intelligence, ML—machine learning, DL—deep learning, BC—bladder cancer, CPATH—computational pathology, H&E—Hematoxylin and eosin.

**Figure 2 cancers-15-04518-f002:**
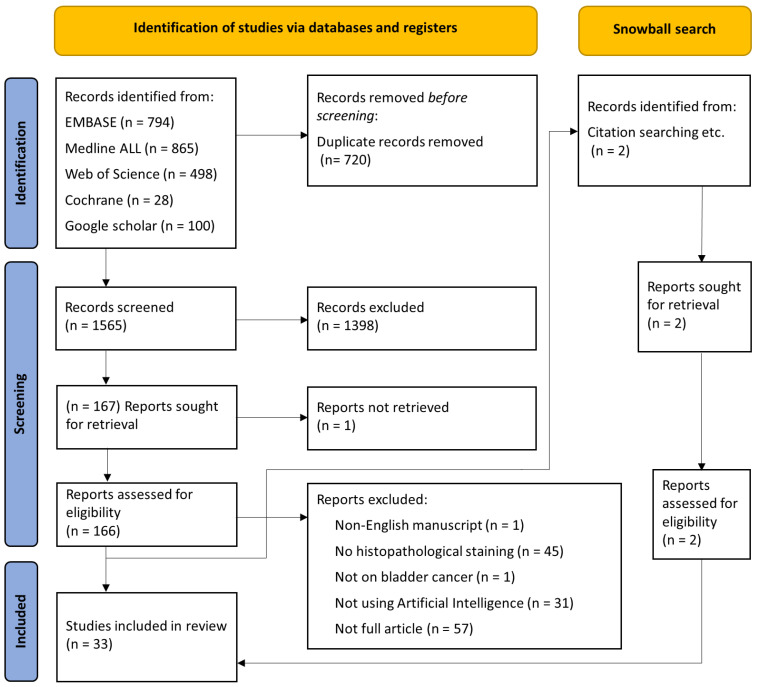
Flowchart of the study according to the PRISMA statement. PRISMA—Preferred Reporting Items for Systematic Reviews and meta-analysis.

## Data Availability

Not applicable.

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
