# Peer review of "Artificial Intelligence in Digital Pathology for Bladder Cancer: Hype or Hope? A Systematic Review"

_cancers, 2023, doi:10.3390/cancers15184518_

Round 1

Reviewer 1 Report

I think the way this manuscript is collected the SOTA literatures that should be improved. More literature should be included. Also, various public dataset/ the issues with those, and also significnat medical relavency with thsoe datasets needed to be discussed. These are missing. 

Author Response

Reviewer 1:

Comments and Suggestions for Authors

“I think the way this manuscript is collected the SOTA literatures that should be improved. More literature should be included. Also, various public dataset/ the issues with those, and also significnat medical relavency with thsoe datasets needed to be discussed. These are missing.”

Point 1.1:I think the way this manuscript is collected the SOTA literatures that should be improved. More literature should be included. More literature should be included.”

Answer: We appreciate your insight regarding the inclusion of recent state-of-the-art (SOTA) literature. For clarity, our literature review was conducted up until March 2022, as mentioned in the manuscript and supplementary.

While our field consistently sees new research and publications, not all of them redefine core principles. After a thorough evaluation, we have ascertained that the recent publications, after our cutoff date, do not introduce transformative changes that would change the conclusions drawn from our established research objectives.

To ensure the inclusivity important recent findings, we added a notable study in the discussion section which was published after our search in The Lancet oncology (lines 514 and 519): " Moreover, in a recent notable study, Wu et al. developed an AI–based model to diagnose lymph node metastases in BC from WSIs which showed superior diagnostic sensitivity over both newer and experienced pathologists. Impressively, its diagnostic ability wasn't limited to BC but extended to breast and prostate cancers as well. With this AI break-through, pathologists are better equipped to identify hard-to-spot micrometastases that could otherwise be overlooked [87] "

Point 1.2: Also, various public dataset/ the issues with those, and also significnat medical relavency with thsoe datasets needed to be discussed.

Answer: Your feedback on the necessity to further explore the public datasets, their inherent issues, and their medical relevance is noted and appreciated. Indeed, while we mentioned the datasets employed in the included studies in Table 1, we recognize that a discussion on the pros and cons of these public datasets could be beneficial for readers. As such, we adjusted the discussion section to elaborate on the advantages and disadvantages associated with the frequently used public datasets in the CPATH BC domain. Additionally, to provide a richer context, we have referred to two recent notable publications that delve into these challenges as described between lines 530 to 549: " The widespread use of The Cancer Genome Atlas (TCGA) dataset, as seen in 42% of the studies included in our review, further underscores the importance of addressing dataset biases. Some models trained on TCGA have shown a tendency to recognize specific institutional patterns, which, although not medically relevant, could unintentionally affect model performance [88, 89]. Moreover, the lack of cross-validation among different cohorts, potential lab-induced tissue artifacts, and the biases from institutional patterns limit model generalizability and clinical application. Elevating the quality and functionality of CPATH in BC demands substantial clinical datasets with adequate clinical follow-up, image noise removal, standardization, and tracking of data collection. Validating models on a broad range of datasets outside of TCGA is essential in upcoming studies to minimize potential biases. Public release of both WSI and annotation datasets, alongside comprehensive disclosure of the algorithm’s outcome and clear decision justifications, remains essential." 

In closing, we sincerely appreciate your insights, which are crucial in elevating our manuscript to the journal's high standards.

Reviewer 2 Report

The systematic review was performed in a good way. English editing is needed. Introduction is complete and clear for the readers. The methodological diagram better explains the used framework, also for non technical readers. Materials and methods is simpler including selection criteria and data synthesis. PRISMA flowchart is included and results show the main insights. Discussion and conclusion are correlated to the study design and objectives. My suggestion is, with the goal to highlight the mission of the research, to better specify the objectives (introduction) and achieved results (results). In this way should be more clear the obtained goal in a bullet points overview. I also suggest to stratify the discussion by endpoint (phenotypes results or outcomes). A table can be a great method in order to show in deep the impact of the review and the potential next steps. This breakdown should add value of the work in terms of potential consideration in featured works in this field. References paragraph is also complete and well done. Attention to the details related to the grammar and form. For example in the table the key "prognosis" must be "Prognosis" as for the Diagnosis key, and so.

Minor editing by native EN is suggested 

Author Response

Reviewer 2:

Comments and Suggestions for Authors:

“The systematic review was performed in a good way. English editing is needed. Introduction is complete and clear for the readers. The methodological diagram better explains the used framework, also for non technical readers. Materials and methods is simpler including selection criteria and data synthesis. PRISMA flowchart is included and results show the main insights. Discussion and conclusion are correlated to the study design and objectives. (1) My suggestion is, with the goal to highlight the mission of the research, to better specify the objectives (introduction) and achieved results (results). In this way should be more clear the obtained goal in a bullet points overview. (2) I also suggest to stratify the discussion by endpoint (phenotypes results or outcomes). A table can be a great method in order to show in deep the impact of the review and the potential next steps. This breakdown should add value of the work in terms of potential consideration in featured works in this field. References paragraph is also complete and well done. Attention to the details related to the grammar and form. For example in the table the key "prognosis" must be "Prognosis" as for the Diagnosis key, and so.”

Point 2.1: “English editing is needed”

Answer: We appreciate your observation and following your suggestion, we thoroughly revised our manuscript to improve the readability and the grammar.

Point 2.2: “My suggestion is, with the goal to highlight the mission of the research, to better specify the objectives (introduction) and achieved results (results). In this way, it should be more clear the obtained goal in a bullet points overview. … A table can be a great method in order to show in-depth the impact of the review and the potential next steps.”

Answer: Thank you for your constructive feedback; we value your emphasis on clarifying the objectives and achieved results of our research. In response to your suggestion, we added detailed bullet points in the introduction to clearly define the goals of our systematic review (lines 105-111): "In this review, we delve into the current status of BC CPATH. Specifically, we:

  • Provide a comprehensive overview of the present BC CPATH landscape in diagnosis and prognosis.
  • Highlight existing gaps in the field.
  • Offer recommendations to address these gaps.
  • Discuss challenges that can shape future research in BC CPATH. ".

Additionally, we acknowledge the importance of the table you highlighted, and thus we have moved what was previously Supplementary Table S2 to the main results section, now labeled as Table 2 on page 13. To offer a clearer perspective on our achieved results, we introduced a section (lines 403 – 408): “In light of our findings, we have identified key gaps in BC CPATH and provided actionable recommendations along with their underlying reasons. Due to the variations in research questions, validation methods, and the WSIs and corresponding annotation databases used, a direct comparison of the studies was not feasible. Nevertheless, Table 2 provides a guide to address the existing challenges and effectively incorporate CPATH into the field of BC diagnosis and prognosis.”

Point 2.3: “I also suggest to stratify the discussion by endpoint (phenotypes results or outcomes).”

Answer: Thank you for your insightful feedback and the suggestion to stratify the discussion based on endpoints. We agree that a more structured approach would greatly enhance the readability and clarity of the discussion.

In response to your recommendation, we have restructured the discussion section to provide a clearer stratification by endpoints. The discussion now comprises the following sections:

  • 1. CPATH for BC diagnosis (line 418)
  • 2. CPATH for BC prediction of prognosis (line 441)
  • 3. Navigating the future: challenges and improvements in BC CPATH (line 504)

Point 2.4: "Attention to the details related to the grammar and form. For example, in the table the key "prognosis" must be "Prognosis" as for the Diagnosis key, and so."

Answer: Thank you for highlighting the grammar and form inconsistencies. We conducted a thorough proofreading, and adjusted the manuscript accordingly. The specific point regarding "prognosis" in the table has been corrected. We appreciate your keen observation.

Reviewer 3 Report

Interesting subject, providing very useful information about AI and its implementation in bladder cancer 

I would suggest to be more specific at the discussion about the more promising approaches to identify the use of AI in order to improve the clinical outcome of the patients.

Author Response

Reviewer 3:

Comments and Suggestions for Authors:

“Interesting subject, providing very useful information about AI and its implementation in bladder cancer

I would suggest to be more specific at the discussion about the more promising approaches to identify the use of AI in order to improve the clinical outcome of the patients.”

Answer:

We are grateful for the reviewer's positive remarks and constructive feedback on our manuscript.

In response to the reviewer’s recommendation to elaborate on the more promising approaches of AI for improving clinical outcomes in bladder cancer, we have enriched our discussion section. Specifically, in lines 543-563, we detailed the significance of CPATH models and their integration within clinical workflows: “To maximize the potential of AI in enhancing patient outcomes, it's essential not just to develop CPATH models but to effectively integrate them within clinical workflows. Continuous monitoring and updating of these models are essential for sustaining their accuracy and reliability. One of the emerging innovations that can help with integrating CPATH in BC clinical practice is fusion models [85]. These models can offer enhanced prediction of prognosis by combining multiple AI techniques and acquiring the unique strengths of individual algorithms.

Patient stratification in clinical trials is essential for their success, which can be augmented by using CPATH’s predictive capabilities. By leveraging CPATH, we can better predict which patients will benefit from specific treatments, thus refining precision medicine and enhancing patient outcomes. An example clinical trial can be more efficient by identifying HR-NMIBC patients who are unlikely to benefit from BCG treatment, allowing for their enrollment in trials investigating other potential treatments; similarly, a subset of MIBC patients who might benefit from immune checkpoints can be identified. CPATH can pinpoint these patient groups. Additionally, the adoption of explainable AI offers transparency into algorithmic decision-making, facilitating clinicians' understanding and confidence in these tools. Importantly, the efficacy of CPATH models depends on diverse datasets, ensuring adaptability across patient groups. Thus, a comprehensive data foundation enhances personalized care for each individual. Beyond its technical innovation, CPATH is directing the BC field towards a paradigm shift in patient care—when CPATH meets BC, it is hope that prevails, not the hype.”

We highlighted promising AI approaches to identify the use of AI to improve clinical outcome prediction. Our revised section aims to provide a comprehensive overview of how CPATH can revolutionize patient care in bladder cancer.

Reviewer 4 Report

The authors present a systemic review of AI research and applications for bladder cancer (BC) pathology. Computational pathology is a rapidly advancing area in various medical domains; therefore, a reflection of AI developments in BC is important to summarize the state of the art. In general, the review is well-written and comprehensive (except the publication time coverage). The Introduction should be edited for better clarity of AI method definitions and concepts. The review also presents issues and possible solutions relevant to all computational pathology domains.

The manuscript presents value for publication by raising awareness of ML models in BC pathology. Nevertheless, I have some remarks:

Major:

1.       The review includes publications before March 2022, which seems a bit outdated in August 2023. Can the review be extended?

2.       Authors refer to the Paige Prostate system; there are also other commercially available and clinically validated tools for prostate. It would be fair to make other appropriate references too or consider removing the citation since the cited application is not related to BC.

3.       The definitions of machine learning (DL) in 1.1 section are confusing. What is “classical machine learning”, line 50?  Then, “ML is a subfield of AI and can improve … without the need for explicit feature engineering”. Then, “The feature engineering part often exploits human knowledge…. Finally, Figure 1A implies a neural network as a necessary(?) step of ML.

4.       Similarly, Figure 2 implies that neural network is a necessary and the only method of data processing (true for DL but not ML). Could Fig 1 and 2 be combined? They are a bit redundant and misleading now.

Minor:

5.       Expressions “…prognosis prediction” in lines 14 and 16: while “prediction of prognosis” is meaningful, it may be confusing for medical audience which uses concepts of prognostic and predictive features of disease.

6.       Line 121: “(H&E) scanned slide image analysis) or (Immunofluorescence staining image analysis)…” - immunohistochemistry not mentioned, although cited later, e.g. line 186.

7.       Lines 187-189: “Certain characteristics of cell nuclei, such as 187 alterations in polarization and shape (e.g., loss of roundness), have been found to correlate 188 with poorer clinical outcomes.” – no reference cited. Is it Loukas et al (segmentation of microvessels)?

8.       Lines 311-313: Needs rephrasing: “Developed algorithm using the combined data predicted 5-y OS with 81%, which performed better than current risk stratification systems based on clinicopathological characteristics.”

9.       Section 3.3.2. Detection of Molecular Alterations includes Ki67 proliferation index and TILs, does not fit the section title (edit the title or make a separate section).

10.   Lines 433-434: Needs rephrasing and a reference: “One included ML-based study achieved 96% accuracy in distinguishing Ta vs. T1 stages.”

11.   Lines 470-471: Needs rephrasing and a reference: “In a notably included study that used DL-based method, molecular subtypes were predicted with 75% accuracy.]. Also, is 75% accuracy is a sufficient prediction level for clinical applications?

Author Response

Reviewer 4:

Comments and Suggestions for Authors:

“The authors present a systemic review of AI research and applications for bladder cancer (BC) pathology. Computational pathology is a rapidly advancing area in various medical domains; therefore, a reflection of AI developments in BC is important to summarize the state of the art. In general, the review is well-written and comprehensive (except the publication time coverage). The Introduction should be edited for better clarity of AI method definitions and concepts. The review also presents issues and possible solutions relevant to all computational pathology domains.

The manuscript presents value for publication by raising awareness of ML models in BC pathology. Nevertheless, I have some remarks:

Major:

  1. The review includes publications before March 2022, which seems a bit outdated in August 2023. Can the review be extended?
  2. Authors refer to the Paige Prostate system; there are also other commercially available and clinically validated tools for prostate. It would be fair to make other appropriate references too or consider removing the citation since the cited application is not related to BC.
  3. The definitions of machine learning (DL) in 1.1 section are confusing. What is “classical machine learning”, line 50? Then, “ML is a subfield of AI and can improve … without the need for explicit feature engineering”. Then, “The feature engineering part often exploits human knowledge…. Finally, Figure 1A implies a neural network as a necessary(?) step of ML.
  4. Similarly, Figure 2 implies that neural network is a necessary and the only method of data processing (true for DL but not ML). Could Fig 1 and 2 be combined? They are a bit redundant and misleading now.

Minor:

  1. Expressions “…prognosis prediction” in lines 14 and 16: while “prediction of prognosis” is meaningful, it may be confusing for medical audience which uses concepts of prognostic and predictive features of disease.
  2. Line 121: “(H&E) scanned slide image analysis) or (Immunofluorescence staining image analysis)…” - immunohistochemistry not mentioned, although cited later, e.g. line 186.
  3. Lines 187-189: “Certain characteristics of cell nuclei, such as 187 alterations in polarization and shape (e.g., loss of roundness), have been found to correlate 188 with poorer clinical outcomes.” – no reference cited. Is it Loukas et al (segmentation of microvessels)?
  4. Lines 311-313: Needs rephrasing: “Developed algorithm using the combined data predicted 5-y OS with 81%, which performed better than current risk stratification systems based on clinicopathological characteristics.”
  5. Section 3.3.2. Detection of Molecular Alterations includes Ki67 proliferation index and TILs, does not fit the section title (edit the title or make a separate section).
  6. Lines 433-434: Needs rephrasing and a reference: “One included ML-based study achieved 96% accuracy in distinguishing Ta vs. T1 stages.”
  7. Lines 470-471: Needs rephrasing and a reference: “In a notably included study that used DL-based method, molecular subtypes were predicted with 75% accuracy.]. Also, is 75% accuracy is a sufficient prediction level for clinical applications?”

Point 4.1:The review includes publications before March 2022, which seems a bit outdated in August 2023. Can the review be extended?”

Answer: We would like to thank the reviewer for the keen observation regarding the publication timeframe.

While our field consistently sees new research and publications, not all of them redefine core principles. After a thorough evaluation, we have ascertained that the recent publications after our cutoff date, do not introduce transformative changes that would recalibrate the conclusions drawn from our established research objectives.

In a testament to our commitment to capturing significant developments, we have highlighted a key article from The Lancet Oncology that was published after our search. As noted between lines 514 to 519: " Moreover, in a recent notable study, Wu et al. developed an AI–based model to diagnose lymph node metastases in BC from WSIs which showed superior diagnostic sensitivity over both newer and experienced pathologists. Impressively, its diagnostic ability wasn't limited to BC but extended to breast and prostate cancers as well. With this AI breakthrough, pathologists are better equipped to identify hard-to-spot micrometastases that could otherwise be overlooked [87]."

Point 4.2:Authors refer to the Paige Prostate system; there are also other commercially available and clinically validated tools for prostate. It would be fair to make other appropriate references too or consider removing the citation since the cited application is not related to BC.”

Answer: We would like to thank the reviewer for the keen observation and constructive feedback. We believe that highlighting the commercial capabilities and regulatory approvals, such as the FDA and CE is pivotal, as it underscores the real-world translational potential and clinical acceptance of such technologies. With this in mind, we have chosen not to remove this section. To emphasize CPATH's clinical potential, we broadened our scope, citing key studies, noting one FDA-approved methodology, and referencing a clinical trial on BC (Lines 84-86 and 501-504):

“Extending beyond bladder-centric, several studies have showcased promising results in leveraging CPATH for improved diagnosis and prognosis prediction, with one method-ology received FDA approval [15-19]. This trend is mirrored in bladder cancer, where there is currently one clinical trials focusing on computational pathology applications (NCT05825950).”

“In several cancer types such as prostate, colorectal, and breast, the potential of CPATH in improving diagnosis and prediction of prognosis has been proven by multiple notable studies which one of the methodologies getting FDA approval [15-19]. In BC, this trend is reflected with a present clinical trial focusing on CPATH approach (NCT05825950).”

Point 4.3:The definitions of machine learning (DL) in 1.1 section are confusing. What is “classical machine learning”, line 50?  Then, “ML is a subfield of AI and can improve … without the need for explicit feature engineering”. Then, “The feature engineering part often exploits human knowledge…. Finally, Figure 1A implies a neural network as a necessary(?) step of ML.

Answer: We would like to thank the reviewer for pointing out the ambiguities in our definitions concerning machine learning. We have adjusted the mentioned section (lines 48-60) in the manuscript to provide more clear and precise explanations. Additionally, we adjusted Figure 1 to better illustrate the processes involved in machine learning and deep learning, thereby clarifying their differences more distinctly.

“AI systems use algorithms, which are mathematical sequences of well-defined instructions, to mimic human-like decision-making processes. The two most commonly used techniques in AI are machine learning (ML) and deep learning (DL). Both aim to enable computers to learn from data but through different approaches (Figure 1). ML, a subfield of AI, needs a phase, known as feature engineering where specific features are manually extracted from data, particularly images, leveraging image processing techniques and human expertise (Figure 1B). This human-driven process distills relevant information from data to shape the ML model’s understanding. Next, the ML model can be trained in a supervised manner, using a set of features with known diagnosis (e.g., stage/grade) or known clinical output (e.g., recurrence/progression) to classify input data into similar groups. While ML often needs resource-intensive feature engineering and domain expertise in its design phase, DL, a subfield of ML, bypasses this by directly extracting relevant features from raw data itself (Figure 1 B).”

Point 4.4:Similarly, Figure 2 implies that neural network is a necessary and the only method of data processing (true for DL but not ML). Could Fig 1 and 2 be combined? They are a bit redundant and misleading now.

Answer: Thank you for your insightful feedback. We merged Figures 1 and 2 (page 3) to provide a more comprehensive illustration and avoid redundancy and potential misconceptions.

Point 4.5: “Expressions “…prognosis prediction” in lines 14 and 16: while “prediction of prognosis” is meaningful, it may be confusing for medical audience which uses concepts of prognostic and predictive features of disease.”

Answer: We appreciate the reviewer's attention to the terminology and its potential impact on the clarity of our manuscript. In light of your suggestion, we have revised the manuscript and replaced "prognosis prediction" with "prediction of prognosis" to ensure clarity for our readers. All these changes are highlighted in green in the manuscript.

Point 4.6: “Line 121: “(H&E) scanned slide image analysis) or (Immunofluorescence staining image analysis)…” - immunohistochemistry not mentioned, although cited later, e.g. line 186.”

Answer: We understand the reviewer's observation regarding the omission of "immunohistochemistry" in line 121 despite its citation in subsequent parts of the manuscript. We would like to thank the reviewer for pointing out this oversight. To ensure comprehensive coverage in our search and to prevent overlooking relevant studies, we intentionally kept our inclusion criteria broad. This encompassed the terms of "bladder cancer" and "artificial intelligence," as well as any terms that are relevant or analogous to them. A detailed list of the exact search terms utilized for each database can be found in Supplementary Table S1. Our focus was on studies using AI to analyze histopathological images. To this end, we applied an exclusion criterion of "No histopathological staining," which is mentioned further in section "2.2. Study Eligibility and Selection." By adopting this approach, we ensured the inclusion of studies that used either immunohistochemistry or H&E staining techniques. We have made the necessary adjustments to the mentioned section for clarity (lines 122-123):

“We employed the specified terms: (bladder cancer) and (artificial intelligence) for study inclusion, along with any terms relevant or analogous to them”

Point 4.7:Lines 187-189: “Certain characteristics of cell nuclei, such as 187 alterations in polarization and shape (e.g., loss of roundness), have been found to correlate 188 with poorer clinical outcomes.” – no reference cited. Is it Loukas et al (segmentation of microvessels)?”

Answer: Thank you for your attentive observation regarding the missing citation. We added the reference as suggested (lines 190-192):

“Certain characteristics of cell nuclei, such as alterations in polarization and shape (e.g., loss of roundness), have been found to correlate with worse clinical outcomes [29].”

Point 4.8: “Lines 311-313: Needs rephrasing: Developed algorithm using the combined data predicted 5-y OS with 81%, which performed better than current risk stratification systems based on clinicopathological characteristics.”

Answer: We would like to thank the reviewer for pointing out the ambiguity in that sentence. We have revised the mentioned sentence for clarity (lines 312-315): “The developed algorithm in this study, using combined data, achieved an 81% accuracy in predicting 5-year overall survival (OS) and outperformed current risk stratification sys-tems based on clinicopathological characteristics.”

Point 4.9: “Section 3.3.2. Detection of Molecular Alterations includes Ki67 proliferation index and TILs, does not fit the section title (edit the title or make a separate section).”

Answer: We greatly appreciate the reviewer’s precise observation concerning the alignment of the section title with its associated content. In response to your feedback, we modified the subsection title to "Detection of Biomarkers and Molecular Alterations" (line 345).

Point 4.10: “Lines 433-434: Needs rephrasing and a reference: One included ML-based study achieved 96% accuracy in distinguishing Ta vs. T1 stages.”

Answer: We are grateful for the reviewer’s acute observation concerning lines 433-434. We adjusted the sentence for more clarity and added the corresponding reference  (lines 451-452): “The mentioned study that highlighted the importance of desmoplastic reaction in BC, used an ML-based method to differentiate between Ta and T1 stages by analyzing H&E WSIs which reached 96% accuracy [37].”

Point 4.11:Lines 470-471: Needs rephrasing and a reference: In a notably included study that used DL-based method, molecular subtypes were predicted with 75% accuracy.]. Also, is 75% accuracy is a sufficient prediction level for clinical applications?”

Answer: We acknowledge and are thankful for the reviewer's keen observation and effort in improving our manuscript. We adjusted the corresponding section to highlight the challenges posed by tumor heterogeneity in BC and elaborate on the significance of the study's findings, especially in the context of molecular subtype detection from histopathological images (lines 488-495):

“In a notably included study on BC that used DL-based method, molecular subtypes were predicted with 75% accuracy and 0.89 AUC [4]. While the performance level has not reached clinical practice standards, it remains significant given the tumor heterogeneity in BC, where diverse genetic and phenotypic profiles within one tumor can challenge classifications. Their method's ability to detect molecular subtypes from histopathological images, without intensive molecular assessment, can pave the way for more efficient diagnostic methods. Moreover, the importance of detecting molecular subtypes is heightened by their clinical relevance.”

Moreover, we underscored the clinical relevance of these subtypes and the correlations identified between morphological features and molecular subtypes. We trust these modifications address your concerns.

Round 2

Reviewer 1 Report

After my suggestion was to be rejected, the authors tried to convince me. Thanks for that. However, it needs further improvements.